# Safety and Efficacy of SGLT2 Inhibitors: A Multiple-Treatment Meta-Analysis of Clinical Decision Indicators

**DOI:** 10.3390/jcm10122713

**Published:** 2021-06-19

**Authors:** Vicente Martínez-Vizcaíno, Ana Díez-Fernández, Celia Álvarez-Bueno, Julia Martínez-Alfonso, Iván Cavero-Redondo

**Affiliations:** 1Health and Social Care Research Center, Universidad de Castilla-La Mancha, 16071 Cuenca, Spain; vicente.martinez@uclm.es (V.M.-V.); celia.alvarezbueno@uclm.es (C.Á.-B.); julia.martinezalfonso@gmail.com (J.M.-A.); ivan.cavero@uclm.es (I.C.-R.); 2Faculty of Health Sciences, Universidad Autónoma de Chile, Talca 3460000, Chile; 3Universidad Politécnica y Artística del Paraguay, Asunción 001518, Paraguay; 4Department of Family and Community Medicine, Hospital La Princesa/Centro de Salud Daroca, 28006 Madrid, Spain; 5Rehabilitation in Health Research Center (CIRES), Universidad de las Americas, Santiago 72819, Chile

**Keywords:** sodium-glucose cotransporter 2 inhibitors, meta-analysis, efficacy, adverse effects, cardiovascular disease, mortality

## Abstract

To jointly assess the safety and effectiveness of sodium-glucose cotransporter 2 inhibitors (SGLT2i) on cardiorenal outcomes and all-cause mortality in type 2 diabetes mellitus (T2DM) with or at high risk of cardiovascular disease (CVD). We performed a systematic review and network meta-analysis, systematically searching the MEDLINE, EMBASE, Cochrane Central Register of Controlled Trials and Web of Science databases up to September 2020. Primary outcomes were composite major adverse cardiovascular events (MACEs), hospitalization for heart failure, all-cause mortality and a composite renal outcome. We performed a random effects network meta-analysis estimating the pooled hazard ratio (HR), risk ratio and number needed to treat (NNT). Six trials evaluating empagliflozin, canagliflozin, dapagliflozin and ertugliflozin met the inclusion/exclusion criteria, which comprised 46,969 patients, mostly with established CVD. Pooled estimates (95% CI) of benefits of SGLT2i in terms of HR and NNT were as follows: for all-cause mortality, 0.85 (0.75, 0.97) and 58 (28, 368); for MACE, 0.91 (0.85, 0.97) and 81 (44, 271); for hospitalization for heart failure, 0.70 (0.62, 0.78) and 32 (20, 55); and for composite renal outcome, 0.61 (0.50, 0.74) and 20 (11, 44). Pooled estimates for serious adverse events were 0.92 (95% CI 0.89, 0.95). In patients with T2DM at cardiovascular risk, ertugliflozin is a less potent drug than empagliflozin, canagliflozin or dapagliflozin to prevent cardiorenal events and all-cause mortality. In addition, our data endorse that empagliflozin is the best treatment option among SGLT2i for this type of patient, but the evidence is not consistent enough.

## 1. Introduction

Sodium-glucose cotransporter 2 inhibitors (SGLT2i) constitute an emerging family of glucose-lowering agents whose mechanism of action is based on the reabsorption of glucose via inhibition of sodium-glucose cotransporter 2 in the proximal convoluted renal tubule [1]. The result is an increase in glycosuria and natriuresis that brings a broad range of metabolic benefits, such as a decrease in glycosylated hemoglobin (HbA1c), body weight, blood pressure and albuminuria [2]. There is enough evidence to state that SGLT2i are a class of antidiabetic oral agents as effective in reducing plasma glucose levels as other available drug options [3]. Likewise, network meta-analyses (NMA) reported no substantial differences in HbA1c reductions among drugs in the SGLT2i family [4,5]. However, beyond glycemic control, evidenced even in animal models [6], the noninferiority randomized controlled trials required for marketing have proven that SGLT2i not only do not increase the risk of cardiovascular or renal events but also reduce the risk of atherosclerotic CVD events and improve kidney outcomes and all-cause mortality [7]. However, the magnitude of these benefits differed by the type of SGLT2i, and serious adverse events have been described that threaten the tolerability of this type of drug, including genital and urinary tract infections, diabetic ketoacidosis, bone fractures, foot and leg amputations and cancer [8]. 

Several mechanism(s), most of them proven in both human and animal models, have been proposed as responsible for the benefits on cardiovascular risk of SGLT2 inhibition [2], including diuresis/natriuresis that lead to blood pressure reduction, enhanced cardiac energy metabolism, reduction of inflammation reduction, inhibition of the sympathetic nervous system, prevention of ischemia/reperfusion injury, decreasing epicardial fat mass, decreasing oxidative stress and improvements on vascular function, among others [2,9,10] (Figure 1).

The implementation of patient-centered and shared decision-making strategies in the care of patients with diabetes has raised the need to personalize treatment strategies and targets [11]. The control of glycemic levels remains the main therapeutic target in the management of patients with T2DM, but this should be framed in the context of a comprehensive cardiovascular and renal risk approach. However, the available information for the shared decision by both physicians and patients about the most advisable therapeutic option among SGLT2i in patients with T2DM at high CVD risk has not yet been synthesized in a single publication. The number needed to treat (NNT) is an absolute effect measure that summarizes the beneficial and harmful effects of medical interventions, which is understandable for patients and practitioners [12] and is appropriate to comparatively synthesize the evidence on the effectiveness and safety of several therapeutic options in meta-analyses and NMAs [13].

The present study aimed to comparatively jointly combine, through a network meta-analysis, survival and NNT data about efficacy and safety of the existing large-scale placebo-controlled trials testing the effect of SGLT2i on cardiovascular and renal outcomes in patients with T2DM who had or were at risk of atherosclerotic CVD.

## 2. Materials and Methods

This systematic review and network meta-analysis is reported according to the Preferred Reporting Items for Systematic Review incorporating Network Meta-analysis (PRISMA-NMA) [14] (Appendix A) and the Cochrane Collaboration Handbook [15]. The study protocol was registered in PROSPERO (registration number: CRD42020178302).

### 2.1. Data Sources and Searches

We conducted a systematic search in the MEDLINE (via PubMed), EMBASE, Cochrane Central Register of Controlled Trials and Web of Science databases from their inception until September 2020. The search strategy combined relevant terms related to (a) SGLT2i type; (b) main outcomes; and (c) type of population (Appendix A). The literature search was complemented by reviewing citations and protocols of the articles considered eligible for the systematic review.

### 2.2. Study Selection

The criteria for including studies were as follows. (i) Randomized clinical trials in any stage, in which the main outcome was to evaluate cardiovascular and renal outcomes. (ii) Studies that included T2DM patients aged 18 years or older without any restriction of age, race, sex or age at diabetes diagnosis who had any established CVD, multiple cardiovascular risk factors and/or renal impairment. (iii) Studies that included at least one SGLT2i in monotherapy or add-on therapy with any approved agent for the treatment of T2DM and compared with a placebo. (iv) Trials with at least one year (52 weeks) of follow-up. (v) Studies that reported incidence of the classical composite three-point major adverse cardiovascular event (MACE) (cardiovascular death, myocardial infarction and stroke), hospitalization for heart failure, all-cause mortality and a composite renal outcome, including combinations of worsening estimated glomerular filtration rate (eGFR) or creatinine, initiation of kidney replacement therapy or transplant, kidney death or CV death. (vi) We did not consider any limitation for selecting studies regarding the language in which they were published. Studies reporting only subsample analysis results were excluded. The literature search was independently conducted by two reviewers (A.D.-F. and I.C.-R.), and disagreements were solved by consensus or involving a third researcher (V.M.-V.).

### 2.3. Data Extraction and Quality Assessment

The following data were extracted from the included studies: (i) clinical trial name; (ii) type of SGLT2i used; (iii) sample size and drug doses; (iv) sponsor; (v) disease status; (vi) length of follow-up; (vii) baseline patient characteristics (age, eGFR, HbA1c and established CVD (%); (viii) event rates (all-cause mortality, MACE, hospitalization for heart failure, composite renal outcome and serious adverse events); and (ix) acceptability (completed alive, non-completed).

The methodological quality of RCTs was evaluated using the Cochrane Collaboration’s tool for assessing risk of bias (RoB2) [16]. This tool includes six domains: selection bias, performance bias, detection bias, attrition bias, reporting bias, and other bias. Each domain was rated as strong, moderate or weak, such that the risk of bias of each study was finally classified as low (with no weak ratings), moderate (with one weak rating) or high (with two or more weak ratings). Furthermore, design, analysis strategy and outcome measures were checked on clinicaltrials.gov registration to assess the bias of adherence to the study protocol.

Both data extraction and quality assessment were independently performed by two reviewers (A.D.-F. and I.C.-R.), and there were no inconsistencies that needed to be solved by a third researcher.

The Grading of Recommendations, Assessment, Development and Evaluation (GRADE) tool was used to assess the quality of the evidence and make recommendations [17]. Each outcome was scored as high, moderate, low or very low evidence, depending on the study design, risk of bias, inconsistency, indirect evidence, imprecision and publication bias.

The included clinical trials were summarized qualitatively in an ad hoc table displaying the effect estimates of each SGLT2i.

### 2.4. Data Synthesis and Analysis

#### 2.4.1. Efficacy of the Main Outcomes and Risk of Adverse Events

We calculated the estimated effect of SGLT2i (empagliflozin, canagliflozin, dapagliflozin and ertugliflozin) relative to placebo using the hazard ratio (HR) for the incidence of each main outcome (all-cause mortality, MACE, hospitalization for heart failure and composite renal outcome), and because survival analyses were not reported, we estimated the risk ratio (RR) for the main adverse events (serious adverse event [18], confirmed hypoglycemic episodes, event consistent urinary tract infection, event consistent with male genital infection, event consistent with volume depletion, acute kidney injury, diabetic ketoacidosis and fracture). For each RCT and outcome, we calculated the number needed to treat (NNT) to prevent an outcome using the reported HR and incidence density; because most studies did not report survival analysis for adverse events, the NNT to avoid an adverse event during the study follow-up was calculated using the absolute risk reduction [19,20,21]. The DerSimonian-Laird random effects method was used to compute the pooled HR, RR and NNT estimates [22]. Forest plots were used to graphically depict the pooled HR for each main outcome and RR for each main adverse event in each treatment comparison. In addition, the pooled NNT estimates along with their confidence intervals (CIs) were included [13]. Positive NNT values indicate the number needed to observe a benefit and negative values indicate the number needed to observe harm.

#### 2.4.2. Clinical Decision Analysis

Comprehensive scatterplots for clinical comparisons were performed. Four-axis scatterplots displaying the HR and NNT for each main outcome (x-axes) and the RR and NNT for serious adverse events (y-axes) by SGLT2i drug versus placebo were generated [13].

#### 2.4.3. Network Meta-Analysis

As noted above, we conducted our NMA according to the PRISMA-NMA statement, under a frequentist perspective, to determine the relative ranking of treatments and the comparative evaluation of the intervention effect as follows:Relative rankings of treatments. Once we comparatively estimated the effectiveness of the different SGLT2i types, the next step was to rank the treatments to identify superiority. The probability that each SGLT2i type was the most effective was presented graphically using rankograms [23]. Additionally, the surface under the cumulative ranking (SUCRA) was estimated for each intervention. SUCRA involves the assignment of a numerical value between 0 and 1 to simplify the classification of each intervention in the rankogram. The best intervention would obtain a value for SUCRA close to 1, and the worst intervention would be a value close to 0 [24].A comparative network meta-analysis of the SGLT2i effect versus placebo is displayed through league tables [25].Sensitivity analysis and small study effect. Sensitivity analyses were conducted to assess the robustness of the summary estimates and to detect whether any particular study represented a large proportion of the heterogeneity. To examine the presence of bias due to the small study effect, a network funnel plot was used to visually scrutinize the criterion of symmetry [26]. All analyses were conducted in Stata 15.0 (Stata, College Station, TX, USA).

## 3. Results

The literature search retrieved 172 studies, of which, after removing duplicates, only 6 studies met the inclusion/exclusion criteria (Figure 2). Appendix A summarizes the characteristics of the studies, including 46,969 patients, whose mean age was between 63.1 and 64.4 years; their eGFR was between 56.2 and 85.2 mL/min/1.73 m2 and their HbA1c was between 8.07% and 8.3%. All the studies were double-blind RCTs with placebo as a control group: two publications of the same study evaluated empagliflozin (doses of 10 and 25 mg) [27,28], two canagliflozin (100 mg, 300 mg or titration from 100 to 300 mg) [29,30], one dapagliflozin 10 mg [31] and one ertugliflozin (5 and 15 mg) [32]. The median follow-up ranged from 2.4 to 4.2 years, and the percentage of patients with established CVD at baseline ranged from 40.6% to 99%. All the RCTs reported data on the following outcomes: all-cause mortality, MACE, hospitalization for heart failure rates and a prespecified composite renal outcome, whose definition differed slightly among studies. The risk of bias, assessed using the Cochrane Collaboration’s tool, was low for all RCTs (Appendix A).

### 3.1. Efficacy of the Main Outcomes

Figure 3 shows the pooled estimates of the effect of SGLT2i relative to placebo in terms of HR and NNT. Overall, all the drugs showed benefits in all outcomes, although they were not always statistically significant. Pooled estimates (95% CI) of benefits of SGLT2i in terms of HR and NNT, respectively, were the following: for all-cause mortality 0.85 (0.75, 0.97) and 58 (28, 368); for MACE, 0.91 (0.85, 0.97) and 81 (44, 271). For hospitalization for heart failure, the HR was 0.70 (0.62, 0.78) and the NNT was settled at 32 (20, 55). Finally, for the composite renal outcome, HR and NNT values were 0.61 (0.50, 0.74) and 20 (11, 44), respectively.

Appendix A summarizes in league tables the pooled HR estimates for the comparison of every SGLT2i using both direct and indirect evidence. In general, it can be noted that for all-cause mortality, comparisons against placebo, the HR estimates (95% CI) for direct and NMA comparisons for empagliflozin were 0.68 (0.57, 0.82) and 0.74 (0.59, 0.92), respectively. For MACE, the HR estimates for both canagliflozin and empagliflozin were 0.74 (0.59, 0.92), but NMA pooled estimates were only significant for empagliflozin (0.84 (0.73, 0.96)). All SGLT2i, compared to placebo, significantly reduced the risk of hospitalization for heart failure in direct and indirect pooled estimates. Finally, the same results were obtained for the composite renal outcome compared to placebo, except for ertugliflozin, whose NMA pooled estimates compared to dapagliflozin and empagliflozin were 1.49 (1.13, 1.99) and 1.45 (1.01, 2.08), respectively.

The quality of evidence for each pairwise comparison according to the GRADE system was rated as high for all of the outcomes (Appendix A).

### 3.2. Risk of Adverse Events 

Significant pooled estimates (95% CI) on adverse event incidence of SGLT2i relative to placebo (Figure 4) in terms of RR and NNT, respectively, were the following: for serious adverse event 0.92 (0.89, 0.95), 40 (29, 68); for events consistent with male genital infection 5.18 (3.22, 8.36) and −54 (−115, −35); and for acute kidney injury 0.77 (0.66, 0.88) and 185 (120, 385).

### 3.3. Clinical Decision Analysis

Figure 5 displays in a four-axis scatterplot, the HR and the NNT for each main outcome (x-axes) and the RR and the NNT for serious adverse events (y-axes). Overall, it can be observed that positions indicating greater efficacy with a lower risk of serious adverse events are those corresponding to the following drugs: empagliflozin for all-cause mortality, empagliflozin and canagliflozin for both MACE and hospitalization for heart failure and dapagliflozin and empagliflozin for composite renal outcome. However, it should be noted that, with the exception of ertugliflozin compared with empagliflozin for all-cause mortality, in all cases, the 95% CI for both efficacy of the risk reduction of the main outcomes and the risk of serious adverse events overlap.

### 3.4. Treatment Ranking

Based on their HR estimates, Figure 6 shows the probabilities of being the most effective drug for the different outcomes of the network meta-analysis. Empagliflozin showed a higher SUCRA (87.6%) for all-cause mortality and MACE (82.7%), canagliflozin for hospitalization for heart failure (77.2%) and dapagliflozin for composite renal disease (83.4%).

### 3.5. Small Study Effects and Publication Bias

There was no evidence of the presence of small-study effects in any outcomes as assessed by funnel plot asymmetry or Egger’s test (Appendix A).

## 4. Discussion

Our study, including data from a recently published RCT assessing the effect of ertugliflozin on cardiovascular and renal outcomes in patients with T2DM and established atherosclerotic CVD [33], not only compares the four marketed SGLT2i in the US and the EU in terms of CVD and renal outcomes and all-cause mortality, but also in terms of serious adverse events. Furthermore, our study comparatively reports the NNT of each drug for every type of outcome (all-cause mortality, MACE, risk of hospitalization for heart failure and a composite renal outcome) and for serious adverse events. Overall, our analysis, contrary to the conclusions of a recent meta-analysis [34], raises doubts on whether, in view of the results of the VERTIS-CV study, we can consider that SGLT2i have a class effect for the prevention of cardiorenal events and all-cause mortality in patients with T2DM with or at high atherosclerotic CVD risk.

Our data endorse that not all SGLT2i are equally effective in preventing cardiovascular events and all-cause mortality (Figure 3), as has been suggested [35]. However, when considering the benefits in the reduction of hospitalization for heart failure and composite renal outcomes, the class effect previously reported for empagliflozin, canagliflozin and dapagliflozin [36] remains consistent after including ertugliflozin data. The results of VERTIS CV suggest that many cardiorenal benefits of SGLT2i go beyond those potentially attributable to improvements in glycemic control, blood pressure or body weight [2], as the benefits of ertugliflozin on these clinical parameters are similar to those reported for the other three drugs, but its benefits in the prevention of CVD events are not. Therefore, it will be necessary to analyze which of the mechanisms underlying the beneficial effects of SGLT2i in the reduction of cardiovascular [37] and renal outcome risk [38,39] are implicated in these effect differences.

The estimates for the adverse events of SGLT2i compared to placebo show consistent benefits of these drugs on the incidence of serious adverse events, i.e., a significantly increased risk of genital infection (in males) and acute kidney injury, and no significant changes in the risk of other commonly reported adverse events, such as hypoglycemia, urinary tract infection, diabetic ketoacidosis, fractures or volume depletion (Figure 4). Overall, despite the increased risk of male genital infection consistently reported in several reviews [40], SGLT2i appear to be an effective strategy to reduce serious adverse events.

An evidence-based clinical decision when prescribing an SGLT2i in patients with T2DM to patients at high risk of atherosclerotic CVD should consider both cardiorenal benefits and safety issues [41]. In this sense, the first conclusion of the analysis of Figure 5 is that ertugliflozin is a less beneficial drug than the other three SGLT2i. The second conclusion, based on the data of the network meta-analysis (Appendix A) and rankograms (Figure 6), along with the pairwise SGLT2i versus placebo comparisons (Figure 3), is that our data, together, reinforce the most accepted current recommendations [42,43] on the management of T2DM in patients with or at risk of atherosclerotic CVD, but exclude ertugliflozin based on these recommendations. Moreover, it can be stated that, in light of the available evidence about risks and benefits, empagliflozin is the SGLT2i most advisable for patients with T2DM who are at high cardiovascular risk. However, as long as there are no studies comparing the effectiveness of reducing the incidence of cardiovascular events of SGLT2i with each other, the soundness of the latter statement is debatable, as all studies from which our pooled data have been calculated were noninferiority trials; therefore, their sample size is not sufficient to demonstrate superiority in the prevention of cardiovascular events. Moreover, the higher proportion of patients with established atherosclerotic CVD in the empagliflozin and ertugliflozin trials casts doubt on the advantages of empagliflozin over canagliflozin and dapagliflozin but not over ertugliflozin.

### Limitations

This study has some limitations to be acknowledged. Although all the samples consisted of patients with established T2DM, differences in inclusion/exclusion criteria, follow-up duration and setting where they were provided health care could influence survival rates [7]. Along the same line of argument, although the convenience of presenting clinically relevant measures does not seem debatable, and presenting NNT for incidence risk differences has been recommended [20], the estimates of this measure of clinical relevance should be cautiously taken into account because they are also influenced by variability in some characteristics of the studies, such as baseline risks, follow-up durations, heterogeneity in outcome definitions or quality of the health care provided by the different clinical settings.

Although data in the league tables and rankograms are quite persuasive, we cannot neglect that the pooled indirect estimates of the NMA lack direct comparisons between the different SGLT2i, which weakens the validity of the NMA conclusions; however, the homogeneity in the clinical profile of the target patients of our study (T2DM with or at risk of atherosclerotic CVD), as well as of the outcomes analyzed, the low risk of bias of the included studies, their not excessively large differences in the follow-up periods and, above all, the consistency between the pairwise comparisons and those of the NMA, endorse that our analyses provide sufficiently solid evidence for decision making in this type of patient. However, because of the lack of data for subgroup analyses in patients with and without metformin treatment at baseline, our data do not provide any evidence about whether metformin may be removed as a first-line treatment for patients with diabetes at high cardiovascular risk, as has been proposed recently [44].

## 5. Conclusions

In conclusion, our study analyzed the efficacy and safety data of SGLT2i in patients with T2DM at high cardiovascular risk and allowed us to conclude that ertugliflozin is a less potent drug than empagliflozin, canagliflozin or dapagliflozin to prevent cardiorenal events and all-cause mortality. In addition, our data endorse that, in light of the available evidence, empagliflozin is the best treatment option among SGLT2i for this population of patients, although for the latter statement, the evidence is not as consistent. Finally, to increase the evidence supporting clinical decision making regarding the best treatment strategy in patients with type 2 diabetes at high cardiovascular risk, it is necessary to conduct studies that compare the efficacy and safety of SGLT2i with each other, as well as studies aimed at elucidating whether metformin should be removed as the first line of treatment in patients with diabetes; moreover, all these studies must include an economic evaluation, including the cost-effectiveness and opportunity cost analyses [45] of each available option.

## Figures and Tables

**Figure 1 jcm-10-02713-f001:**
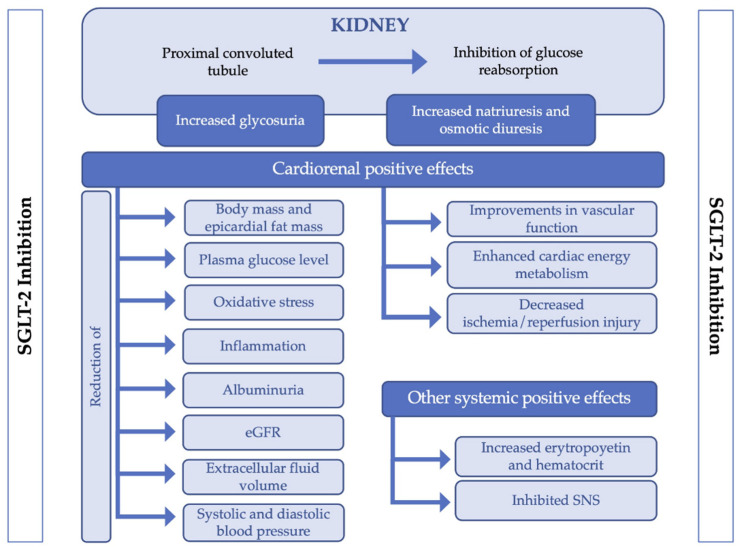
Mechanism of action of SGLT-2 inhibition and related positive effects. Abbreviations: eGFR: estimated Glomerular Filtration Rate; SNS: Sympathetic nervous system.

**Figure 2 jcm-10-02713-f002:**
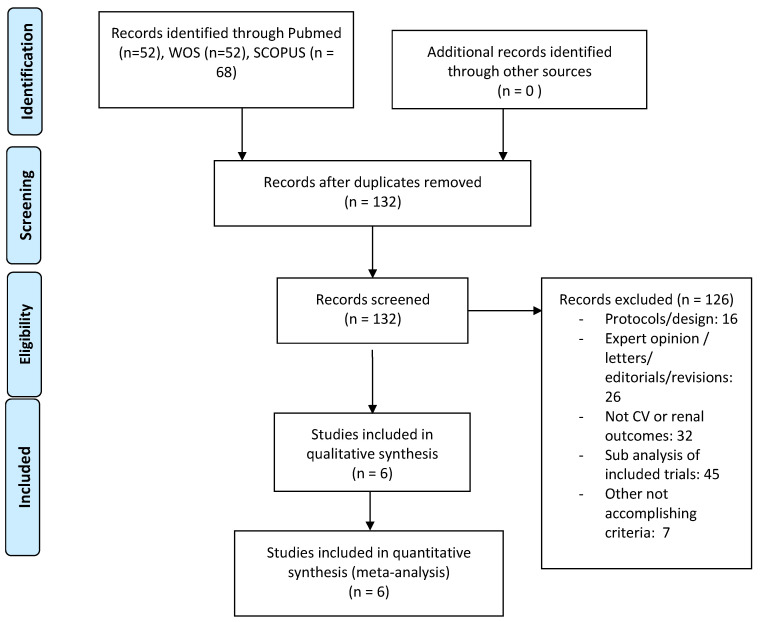
Literature search PRISMA (Preferred Reporting Items for Systematic Reviews and Meta-analyses) consort diagram.

**Figure 3 jcm-10-02713-f003:**
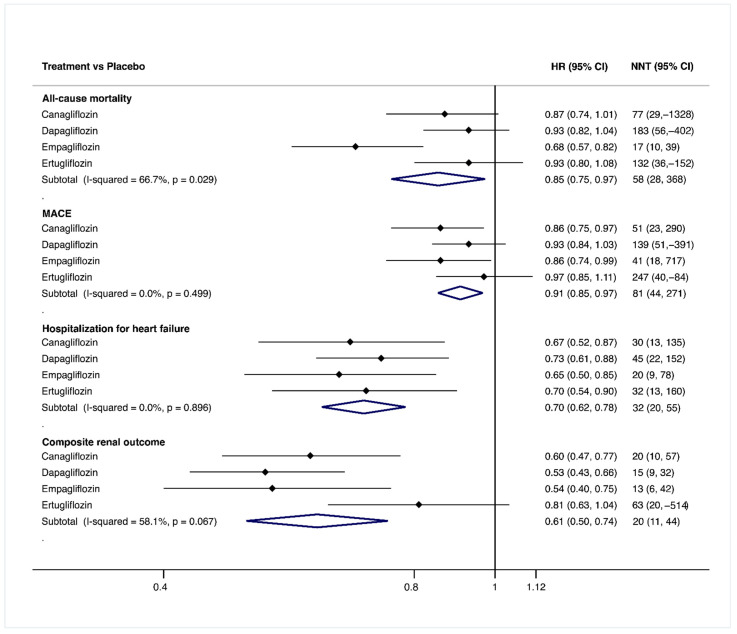
Forest plot for canagliflozin, dapagliflozin, empagliflozin and ertugliflozin vs. placebo to assess the efficacy of the main outcomes. Abbreviations: HR: Hazard Ratio; NNT: Number needed to treat (calculated as 1/Absolute risk reduction (ARR), with ARR = exposed incidence minus non-exposed incidence); CI: Confidence Interval; MACE: Mayor Adverse Cardiovascular Events. The HR and its 95% CI and the corresponding NNT and its 95% CI are shown. HR has been extracted from original studies and has been transformed to NNT. Positive values of NNT means the number needed to observe a benefit and negative values means the number needed to observe harm.

**Figure 4 jcm-10-02713-f004:**
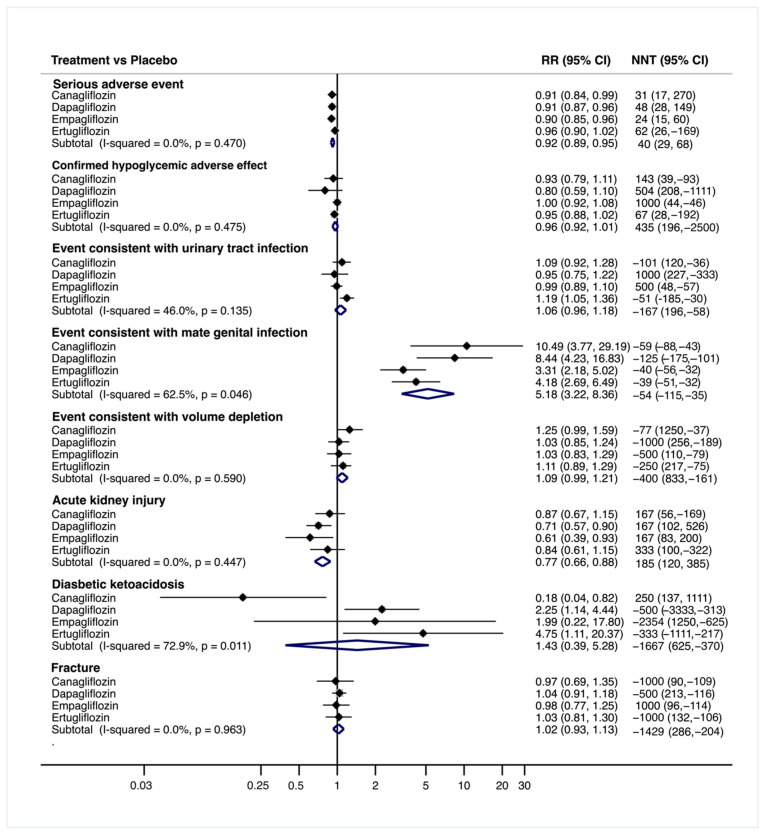
Forest plot for canagliflozin, dapagliflozin, empagliflozin and ertugliflozin vs. placebo to assess adverse effects. Abbreviations: CI: confidence interval; RR: relative risk; NNT: number needed to treat (calculated as 1/Absolute risk reduction (ARR), with ARR = exposed incidence minus non-exposed incidence). The RR and its 95% CI and the corresponding NNT and its 95% CI for each comparison are shown. The RR effect measure has been estimated from a number of events from original studies and has been transformed to NNT. Positive values of NNT means the number needed to observe a benefit and negative values means the number needed to observe harm.

**Figure 5 jcm-10-02713-f005:**
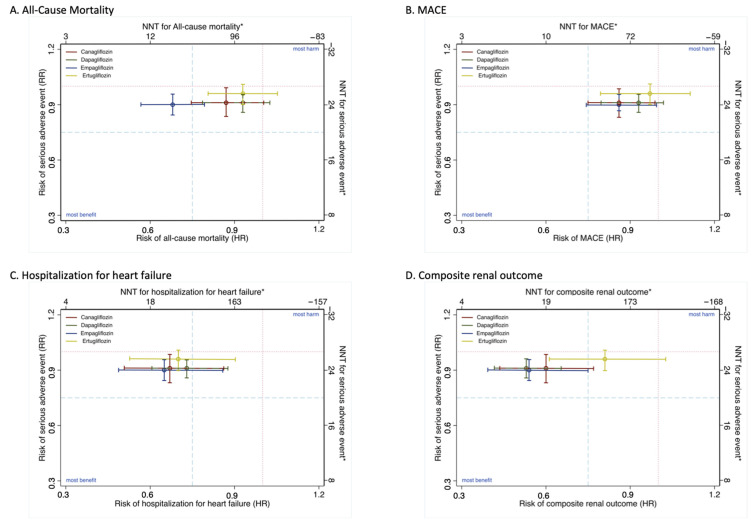
Comprehensive scatterplot for the efficacy of canagliflozin, dapagliflozin, empagliflozin and ertugliflozin vs. placebo for main outcomes (x-axis) and serious adverse events (y-axis). (**A**) All-cause mortality; (**B**) MACE; (**C**) Hospitalization for heart failure and (**D**) Composite Renal. Treatments lying on the lower left-hand side quarter are the most beneficial. Abbreviations: RR: relative risk; HR: hazard ratio; NNT: number needed to treat (calculated as 1/Absolute risk reduction (ARR), with ARR = exposed incidence minus non-exposed incidence). * NNTs have been estimated from the pooled NNTs for each outcome and serious adverse event from Figure 3 and Figure 4.

**Figure 6 jcm-10-02713-f006:**
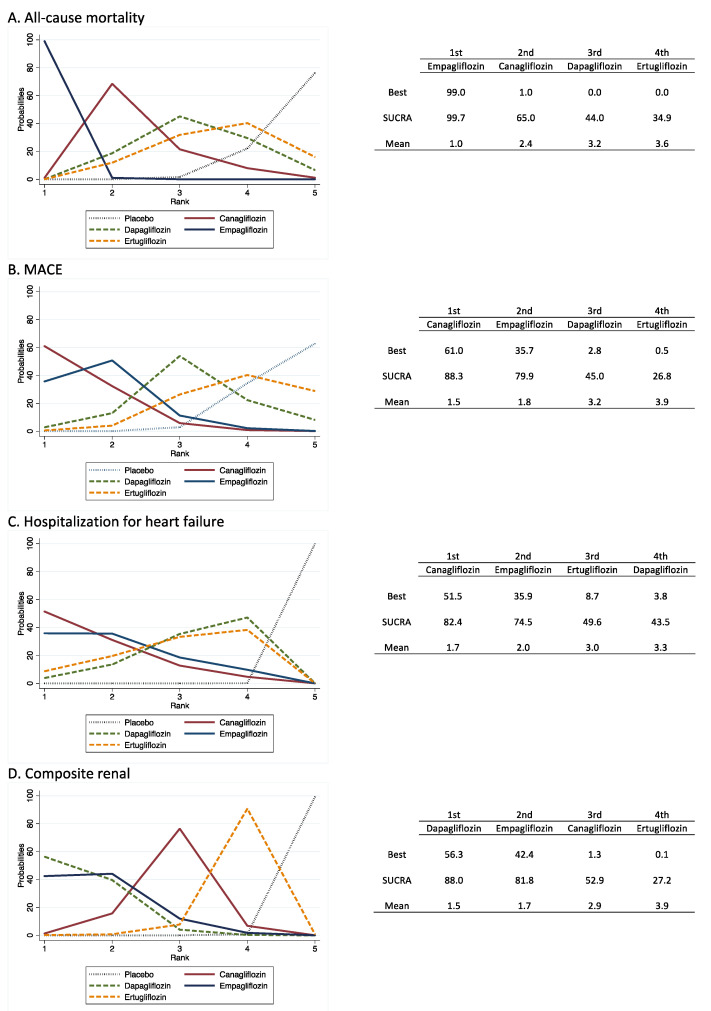
Relative rankings of treatments. (**A**) All-cause mortality; (**B**) MACE; (**C**) Hospitalization for heart failure and (**D**) Composite Renal.

## Data Availability

No new data were created or analyzed in this study. Data sharing is not applicable to this article.

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
