# Peer review of "Safety and Efficacy of SGLT2 Inhibitors: A Multiple-Treatment Meta-Analysis of Clinical Decision Indicators"

_jcm, 2021, doi:10.3390/jcm10122713_

Round 1
Reviewer 1 Report
This is a very interesting study, well written and easy to understand. I don’t have any major comments.
Minor comments:
Abstract: The first sentence is confusing: “… patients with type 2 diabetes at high cardiovascular risk or with CV disease”?
In Figure 1: NNTB and NNTH should be defined.
In Figure 3: ARR should be defined.
Author Response
Authors: We thank the positive feedback of the reviewer. The sentence in the abstract has been clarified and abbreviations have been included, as follows:
“…in type 2 diabetes mellitus (T2DM) with or at high cardiovascular disease (CVD)”.
“Calculated as 1/Absolute risk reduction (ARR), being ARR= exposed incidence minus non-exposed incidence”.
“Positive values of NNT means number needed to benefit (NNTB) and negative values means number needed to harm (NNTH)”.
Reviewer 2 Report
Martinez-Vizcaino et al., have shown a systematic review and network meta-analysis, systematically searching the MEDLINE, EMBASE, Cochrane Central Register of Controlled Trials, and Web of Science databases up to September 2020. Primary outcomes were composite major adverse cardiovascular events (MACEs), hospitalization for heart failure, all-cause mortality and a composite renal outcome. They performed a random effects network meta-analysis estimating the pooled hazard ratio (HR), risk ratio and number needed to treat (NNT). Six trials evaluating empagliflozin, canagliflozin, dapagliflozin and ertugliflozin met the inclusion/exclusion criteria, which comprised 46 969 patients, mostly with established CVD. Pooled estimates (95% CI) of benefits of SGLT2i in terms of HR and NNT were as follows: for all-cause mortality, 0.85 (0.75, 0.97) and 58 (28,368); for MACE, 0.91 (0.85, 0.97) and 81 (44, 271); for hospitalization for heart failure, 0.70 (0.62, 0.78) and 32 (20, 55); and for composite renal outcome, 0.61 (0.50, 0.74) and 20 (11, 44). Pooled estimates for serious adverse events were 0.92 (95% CI 0.89, 0.95). They conclude that in patients with T2DM at cardiovascular risk, ertugliflozin is a less potent drug than empagliflozin, canagliflozin or dapagliflozin to prevent cardiorenal events and all-cause mortality. In addition, they describe that empagliflozin is the best treatment option among SGLT2i for this type of patient.
I have the following comments:
- The introduction should be more complete and with sections including in vivo and in vitro studies.
- A flow chart on the included and chosen studies is necessary.
- A figure about the mechanism of action of the iSGLT-2 is necessary to show.
- The bibliography is scarce and it is necessary to increase it.
Author Response
- The introduction should be more complete and with sections including in vivo and in vitro studies.
Authors: Thank you for the reviewer’s comment. We have included in the introduction some sentences related to in vivo and in vitro studies:
“However, beyond glycemic control, evidenced even in animal models [6]…”
“Several mechanism(s), most of them proven in both human and animal models, have been proposed as responsible for the benefits on cardiovascular risk of SGLT2 inhibition [2], including diuresis/natriuresis that lead to blood pressure reduction, enhanced cardiac energy metabolism, reduction of inflammation, inhibition of the sympathetic nervous system, prevention of ischemia/reperfusion injury, decreasing epicardial fat mass, decreasing oxidative stress and improvements on vascular function [2,9,10] (Figure 1).”
- A flow chart on the included and chosen studies is necessary.
Authors: The flow chart was included in the supplementary material. However, we have moved it in the main text as Figure 2.
- A figure about the mechanism of action of the iSGLT-2 is necessary to show.
Authors: We agree with the reviewer. A figure (Figure 1) and the above paragraph have been included in the introduction section.
- The bibliography is scarce and it is necessary to increase it.
Authors: We appreciate the reviewer’s suggestion. The following references have been included throughout the manuscript:
Ferreira, G.S.; Veening-Griffioen, D.H.; Boon, W.P.C.; Hooijmans, C.R.; Moors, E.H.M.; Schellekens, H.; van Meer, P.J.K. Comparison of drug efficacy in two animal models of type 2 diabetes: A systematicreview and meta-analysis. Eur. J. Pharmacol. 2020, 879, 173153, doi:10.1016/j.ejphar.2020.173153.
Lopaschuk, G.D.; Verma, S. Mechanisms of Cardiovascular Benefits of Sodium Glucose Co-Transporter 2 (SGLT2)Inhibitors: A State-of-the-Art Review. JACC. Basic to Transl. Sci. 2020, 5, 632–644, doi:10.1016/j.jacbts.2020.02.004.
Alshnbari, A.S.; Millar, S.A.; O’Sullivan, S.E.; Idris, I. Effect of Sodium-Glucose Cotransporter-2 Inhibitors on Endothelial Function: ASystematic Review of Preclinical Studies. Diabetes Ther. Res. Treat. Educ. diabetes Relat. Disord. 2020, 11, 1947–1963, doi:10.1007/s13300-020-00885-z.
Margonato, D.; Galati, G.; Mazzetti, S.; Cannistraci, R.; Perseghin, G.; Margonato, A.; Mortara, A. Renal protection: a leading mechanism for cardiovascular benefit in patients treatedwith SGLT2 inhibitors. Heart Fail. Rev. 2021, 26, 337–345, doi:10.1007/s10741-020-10024-2.
Liu, Z.; Ma, X.; Ilyas, I.; Zheng, X.; Luo, S.; Little, P.J.; Kamato, D.; Sahebkar, A.; Wu, W.; Weng, J.; et al. Impact of sodium glucose cotransporter 2 (SGLT2) inhibitors on atherosclerosis: frompharmacology to pre-clinical and clinical therapeutics. Theranostics 2021, 11, 4502–4515, doi:10.7150/thno.54498.
Li, S.; Vandvik, P.O.; Lytvyn, L.; Guyatt, G.H.; Palmer, S.C.; Rodriguez-Gutierrez, R.; Foroutan, F.; Agoritsas, T.; Siemieniuk, R.A.C.; Walsh, M.; et al. SGLT-2 inhibitors or GLP-1 receptor agonists for adults with type 2 diabetes: aclinical practice guideline. BMJ 2021, 373, n1091, doi:10.1136/bmj.n1091.
Round 2
Reviewer 2 Report
No more comments